# Micronutrient Deficiency and Muscular Status in Inflammatory Bowel Disease

**DOI:** 10.3390/nu16213763

**Published:** 2024-11-01

**Authors:** Joonhee Han, Hyun Joo Song, Min Sook Kang, Hogyung Jun, Heung Up Kim, Ki Soo Kang, Donghyoun Lee

**Affiliations:** 1Department of Internal Medicine, Jeju National University Hospital, Jeju National University College of Medicine, Jeju 63241, Republic of Korea; urlsiguna@nate.com (J.H.); junhokyoung@naver.com (H.J.); kimhup@jejunu.ac.kr (H.U.K.); 2Department of Food & Nutrition Service Team, Jeju National University Hospital, Jeju 63241, Republic of Korea; rat0712@naver.com; 3Department of Pediatrics, Jeju National University Hospital, Jeju National University College of Medicine, Jeju 63241, Republic of Korea; kskang@jejunu.ac.kr; 4Department of Surgery, Jeju National University Hospital, Jeju National University College of Medicine, Jeju 63241, Republic of Korea; dlee@jejunuh.co.kr

**Keywords:** micronutrients, muscle, inflammatory bowel disease

## Abstract

Micronutrient deficiencies are common in inflammatory bowel disease (IBD). The aim of this study was to evaluate micronutrient deficiencies and identify muscular status of patients with IBD. From June 2019 to October 2021, a total of 105 patients with IBD were enrolled prospectively. To obtain objective data, micronutrients were measured in the patients' serum, and body composition analysis was performed using bioelectrical impedance analysis. There were 51 patients with ulcerative colitis (UC) and 54 with Crohn’s disease (CD), while the gender ratio (M: F) was 54:51. The average age was 37 ± 18 years, which was significantly lower in patients with CD than UC (29 ± 16 vs. 45 ± 16, *p* < 0.001). Iron and magnesium were lower in patients with CD compared to UC, respectively (63.3 ± 42.5 vs. 82.8 ± 44.0 µg/dL, *p* = 0.024, 2.08 ± 0.15 vs. 2.15 ± 0.19 mg/dL, *p* = 0.036). Vitamin D levels showed insufficiency in patients with UC and deficiency (below 20 ng/mL) in patients with CD (20.1 ± 10.6 vs. 19.0 ± 9.9 ng/mL, *p* = 0.567). In the UC and CD patient groups, skeletal muscle index (SMI) and adjusted skeletal muscle mass were lower in patients with CD compared to UC (SMI: 32.8 ± 4.7 vs. 35.8 ± 5.5%, *p* < 0.004, adjusted skeletal muscle: 7.0 ± 1.5 vs. 8.2 ± 1.9 kg/m^2^, *p* < 0.001). In conclusion, decreased trace elements, specifically iron, magnesium, and vitamin D, as well as skeletal muscle mass were observed to be prominent in patients with CD as compared to UC.

## 1. Introduction

Inflammatory bowel disease (IBD), which includes Crohn’s disease (CD) and ulcerative colitis (UC), is characterized by chronic, recurring inflammation in the gastrointestinal (GI) tract. While CD can affect any part of the GI tract, UC is primarily confined to the rectum and colon [1,2,3]. Although the exact mechanisms are unclear, environmental risk factors such as early-life exposures, lifestyle, hygiene, vaccinations, drug use, GI pathogens, and surgeries are believed to contribute to the development of IBD [3].

Malnutrition, including protein–energy and micronutrient deficiencies, is common among IBD patients [4,5]. Both macro- and micronutrient deficiencies, along with sarcopenia, can significantly impact patients’ quality of life and increase the risk of complications. According to guidelines by the European Working Group on Sarcopenia in Older People and other expert groups, sarcopenia is characterized by low skeletal muscle mass, strength, and/or performance [6,7,8].

Micronutrient deficiencies affect nearly half of IBD patients, with iron, vitamin B12, vitamin K, folic acid, selenium, zinc, vitamin B6, and vitamin B being the most frequently observed [5]. These deficiencies are more prevalent in CD than in UC, especially during periods of high disease activity [4], and can exacerbate disease complications. For example, iron deficiency is closely linked to anemia in severe IBD patients [9].

Bioelectrical impedance analysis (BIA) is a noninvasive method used to assess body composition and muscle mass, providing valuable insights into the physical status of patients [10]. Despite its usefulness, only a limited number of studies have investigated the use of BIA in IBD patients [11,12,13].

In Korea, the incidence of IBD is rising [14,15], with a similar trend observed on Jeju Island. This region presents a unique research opportunity due to its relatively consistent dietary habits and ease of patient follow-up, owing to geographic isolation. Although interest in the nutritional status of IBD patients is growing, there is still a lack of specific data on micronutrient deficiencies in the Korean population. Thus, this study aims to explore the nutritional status, micronutrient deficiencies, and muscle mass of IBD patients.

## 2. Materials and Methods

### 2.1. Study Design

This study is a cross-sectional study that prospectively enrolled patients who received treatment from June 2019 to October 2021 in the Division of Gastroenterology, Departments of Internal medicine and Pediatrics at Jeju National University Hospital.

### 2.2. Participants

The study was conducted on a total of 131 consecutive IBD patients diagnosed at our hospital during the study period. The age, gender, height, weight, family history of IBD, alcohol consumption status, and smoking history of both UC and CD patients were recorded. Medical evaluations included the presence of diabetes, hypertension, hyperlipidemia, disease duration and activity, and the use of 5-aminosalicylic acid (ASA), steroids, immunomodulators, and biological agents. IBD diagnosis was based on clinical, endoscopic, radiologic, and histopathological criteria established by a gastroenterologist or pediatric specialist.

Exclusion criteria were as follows: (i) Patients who were treated or were being treated for IBD prior to the study period; (ii) those who previously underwent gastrointestinal tract resection, which could cause nutrient deficiencies; (iii) patients with a history of cancer or with a stoma; (iv) patients lost to follow-up during the study period; and (v) those with incomplete medical records. A total of 26 patients were excluded based on these criteria.

### 2.3. Patient Assessment Scale and Disease Definition

Disease activity was evaluated by the physician using the Crohn’s Disease Activity Index for CD and the Mayo or Partial Mayo Score for UC. For patients diagnosed in pediatrics, the Pediatric Crohn’s Disease Activity Index and the Pediatric Ulcerative Colitis Activity Index were used. Disease activity was classified as remission, mild, moderate, or severe based on the scores.

Dyslipidemia was determined according to the “Evidence-based Recommendations for Dyslipidemia in Primary Care” by the Korea Disease Control and Prevention Agency. It was defined as meeting one of the following criteria: ① total cholesterol ≥ 240 mg/dL; ② LDL-C ≥ 160 mg/dL; ③ triglycerides ≥ 200 mg/dL; or ④ HDL-C < 40 mg/dL. Alcohol consumption was defined according to the National Institute on Alcohol Abuse and Alcoholism guidelines as drinking 14 g of pure alcohol daily during the study period. Individuals who had abstained before the study were classified as non-drinkers.

### 2.4. Blood Analysis

Biochemical assessment of nutritional status included measurements of hemoglobin, albumin, calcium, iron, ferritin, phosphate, folate, magnesium, serum 25-hydroxy (OH) vitamin D, vitamin B12, and zinc. These levels were compared between CD and UC patients. Most micronutrients were measured using the OLYMPUS AU5421 (Beckman-Coulter, Fullerton, CA, USA). Serum 25-hydroxy (OH) vitamin D and vitamin B12 were measured using immunoassays or liquid chromatography–tandem mass spectrometry. Zinc was assessed using atomic absorption spectrometry

### 2.5. Bioelectrical Impedance Analysis

Total amount of muscle in the body was measured through a widely used body composition analyzer (X-Scan Plus 2^®^, Jawon Medical Co., Ltd., Gyeongsan-si, Korea) that provides a set of raw bioelectrical measurements of Z and Xc, for five parts of the body (both arms and legs and the trunk) in multiple frequencies ranging from 1 kHz to 1000 kHz/ch. The principle of body composition analyzer is as follows: the amount of water is measured by electrical resistance, and the amount of muscle and fat is estimated using a particular formula. Since the number of visceral and cardiac muscles is constant according to age, gender, and weight, the amount of skeletal muscle is calculated through this. Based on this, we calculated two parameters of the skeletal muscle index (SMI, skeletal muscle mass × 100/weight) and adjusted skeletal muscle mass, skeletal muscle mass by the square of the height [16].

### 2.6. Statistical Methods

All statistical analyses were performed using IBM SPSS Statistics version 24.0 (IBM Corporation, Armonk, NY, USA). We analyzed patients’ age, gender, smoking and alcohol status, family history, past medical history, body mass index (BMI), date of IBD diagnosis, disease activity, and medical treatment in terms of averages and proportions. Continuous variables were described as mean ± standard deviation, and comparisons between two groups were made using Student’s *t*-test. Non-continuous variables were analyzed using the chi-square test. All variables, except family history, diabetes, hypertension, and hyperlipidemia, followed a normal distribution. Variables such as family history, diabetes, hypertension, and hyperlipidemia were analyzed using Fisher’s exact test. Linear-by-linear association was used for trend analysis. A *p*-value of <0.05 was considered statistically significant.

### 2.7. Ethics Statement

All the participants provided written informed consent before participating in the study. This study was approved by the Institutional Review Board of Jeju National University Hospital (IRB No. 2019-06-001).

## 3. Results

### 3.1. Demographic Characteristics of Patients

There were 51 patients with ulcerative colitis (UC), and 54 with Crohn’s disease (CD), while the gender ratio (M:F) was 54:51. The average age was 37 ± 18 years, which was significantly lower in patients with CD than UC (29 ± 16 vs. 45 ± 16, *p* < 0.001). The mean BMI was 22.0 ± 3.7 kg/m^2^, lower in patients with CD than UC (21.2 ± 3.8 kg/m^2^, vs. 22.0 ± 3.3 kg/m^2^, *p* = 0.030). Family history of IBD was identified in 4.8% (n = 5, UC 4 vs. CD 1). Most were nonsmokers (84.8%) and current alcohol drinker was at 22.9% (n = 22). For underlying disease, hyperlipidemia was more common in patients with UC vs. CD (15.9% vs. 1.9%, *p* = 0.014). Mean disease duration of IBD was 77.5 ± 68.3 months. Regarding medication, 5-ASA was more commonly prescribed in patients in UC than CD (88.2% vs. 57.4%, *p* < 0.001). In contrast, immune modulators such as azathioprine and 6-MP were more commonly taken in patients in CD than UC (61.1% vs. 19.6%, *p* < 0.001). Biologic agents were more commonly used in patients with CD compared with UC (43.4% vs. 23.5%, *p* = 0.032). The disease activity of the patients was 64.8% in remission, with 17.1% mild, 15.2% moderate, and 2.9% severe (Table 1).

### 3.2. Micronutrient Deficiency

When blood tests were conducted for hemoglobin, total protein, albumin, and calcium, no significant differences were found between patients with UC and CD. However, iron levels were significantly lower in CD patients compared to UC (63.3 ± 42.5 vs. 82.8 ± 44.0 µg/dL, *p* = 0.024). Ferritin, phosphorus, folic acid, vitamin B12, and zinc were within normal limits. Magnesium was also lower in CD patients (2.08 ± 0.15 vs. 2.15 ± 0.19 mg/dL, *p* = 0.036). While vitamin D levels indicated insufficiency in UC and deficiency in CD (20.1 ± 10.6 vs. 19.0 ± 9.9 ng/mL), the difference was not statistically significant (*p* = 0.567, Table 2).

### 3.3. Muscular Status

Bioelectrical impedance analysis indicated that skeletal muscle was lower in patients with CD compared to UC (19.2 ± 5.5 vs. 22.7 ± 6.2. kg, *p* = 0.001). In addition, SMI and adjusted skeletal muscle mass were lower in patients with CD compared to UC (SMI: 32.8 ± 4.7 vs. 35.8 ± 5.5%, *p* < 0.004, adjusted skeletal muscle: 7.0 ± 1.5 vs. 8.2 ± 1.9 kg/m^2^, *p* < 0.001) (Table 3).

## 4. Discussion

IBD is a chronic condition that significantly increases the risk of malnutrition and sarcopenia, ultimately impacting overall patient health [17]. The severity and course of IBD, such as disease activity, duration, and extent, directly influence nutritional status. CD and UC often results in nutrient malabsorption, with CD patients being particularly affected due to the frequent involvement of the small bowel. While nutrient deficiencies tend to be more severe in CD, both CD and UC patients commonly experience deficiencies in essential micronutrients, including iron, magnesium, and vitamins D and B12 [18]. Addressing these deficiencies is crucial to prevent worsening of disease progression and associated complications.

Traditional nutritional assessments, which rely on anthropometric measurements like height, weight, body circumferences, skinfold thickness, and BMI, often fail to provide a complete picture of a patient’s nutritional status, especially regarding muscle mass and fat distribution. However, recent advancements, such as BIA, offer a more comprehensive evaluation of body composition, including skeletal muscle mass and fat reserves.

In this study, the mean BMI was 22.0 ± 3.7 kg/m^2^, with CD patients having a significantly lower BMI than UC patients (21.2 ± 3.8 kg/m^2^ vs. 22.0 ± 3.3 kg/m^2^, *p* = 0.030), consistent with previous research findings. Although micronutrient absorption typically decreases with age due to reduced stomach acid production and altered vitamin D metabolism, our study revealed that malabsorption and nutrient deficiencies were more prevalent in CD patients, despite their significantly younger age compared to UC patients. This suggests that small bowel involvement in CD may have a more substantial impact on micronutrient absorption than age-related decline. A larger-scale study is currently being prepared to control for confounding factors such as age more rigorously.

Sarcopenia in IBD has a multifactorial origin, with vitamin D deficiency being a significant contributor. In our study, vitamin D levels were insufficient in UC patients and deficient in CD patients (20.1 ± 10.6 vs. 19.0 ± 9.9 ng/mL, *p* = 0.567), though the difference was not statistically significant. Recent studies suggest a possible association between vitamin D deficiency and disease activity in CD [19,20]. While these studies are primarily retrospective and observational, they, along with our findings, support the hypothesis that vitamin D supplementation may benefit disease severity and bone health in IBD patients.

A retrospective Korean study reported that nearly 50% of newly diagnosed CD patients had sarcopenia, which is known to be an independent risk factor for surgery and postoperative complications in IBD [21]. In our study, CD patients exhibited significantly lower SMI and adjusted muscle mass than UC patients (SMI: 32.8 ± 4.7 vs. 35.8 ± 5.5, *p* < 0.004; adjusted skeletal muscle: 7.0 ± 1.5 vs. 8.2 ± 1.9, *p* < 0.001). These findings highlight the need for targeted nutritional interventions, especially for CD patients, to mitigate the risk of sarcopenia and its associated complications.

Most participants in this study had relatively favorable nutritional status, with remission and mild cases accounting for 81.9% of the total (64.8% and 17.1%, respectively), while severe cases made up only 2.9%. Among CD patients, one severe case (1.8%) involved high fevers, persistent vomiting, and intestinal obstruction, despite the use of immunomodulators and biological agents. However, CD patients generally had poorer nutritional status as indicated by significantly lower levels of iron (63.3 ± 42.5 vs. 82.8 ± 44.0 µg/dL, *p* = 0.024) and magnesium (2.08 ± 0.15 vs. 2.15 ± 0.19 mg/dL, *p* = 0.036). Although iron deficiency was more frequent in CD than UC, it remained within the normal range, likely due to prolonged iron supplementation (mean disease duration: 77.5 ± 68.3 months).

According to the ESPEN guidelines, iron supplementation is recommended for IBD patients with iron deficiency anemia to normalize hemoglobin levels and iron stores [22]. Vitamin and mineral deficiencies are common in IBD and may contribute to the severity of the disease and related comorbidities. These deficiencies can result from decreased intake, malabsorption, or excess losses. Iron is particularly crucial for blood production, with anemia prevalence ranging between 36 and 76% in IBD patients [9].

Recent systematic reviews indicate that obesity also increases the risk of IBD development (HR: 1.20, 95% CI: 1.08–1.34) [23]. Both underweight and obesity independently raise the risk of CD, although there is no clear association between BMI and UC risk [24]. Currently, a larger proportion of IBD patients are either overweight or obese, emphasizing the need for proper management of obesity-related chronic inflammation [23,25]. Nutritional management, including dietary interventions by dietitians, is essential to address both undernutrition and obesity in IBD patients [26,27,28].

While this study offers valuable insights into the nutritional status of IBD patients, it has certain limitations. Being a single-center with a relatively small sample size, there is a potential for selection bias. Additionally, the study lacked a healthy non-IBD control group for comparison of body composition and micronutrient concentrations. Furthermore, many patients had a long disease duration and were already in remission or mild stages due to effective drug treatments, potentially influencing their nutritional status. The study’s reliance on mean values also limited detailed analysis of individual variations. Lastly, its cross-sectional design restricts the ability to draw causal inferences. Future research should include larger cohorts and more comprehensive dietary intake assessments to explore the relationship between nutrition, body composition, and disease outcomes. Despite these limitations, this study highlights differences in micronutrient deficiencies and muscle condition between CD and UC patients and highlights the need for further large-scale studies.

## 5. Conclusions

This study is meaningful in that it was possible to see differences depending on the patients’ micronutrient deficiency and muscular status according to IBD subtype (CD vs. UC). Decreased trace elements, specifically iron, magnesium, and vitamin D, as well as skeletal muscle mass were observed to be prominent in patients with CD as compared to UC.

## Figures and Tables

**Table 1 nutrients-16-03763-t001:** Demographic characteristics of IBD patients.

Variables	IBD (n = 105)	Ulcerative Colitis (n = 51)	Crohn’s Disease (n = 54)	*p*-Value
Age	37 ± 18 (11–86)	45 ± 16.4	29 ± 16.1	<0.001 *
Sex, female	51 (51.4%)	29 (56.9%)	22 (40.7%)	0.099
BMI (Kg/m^2^)	22.0 ± 3.7	22.8 ± 3.3	21.2 ± 3.8	0.030 *
Family history of IBD	5 (4.8%)	4 (7.8%)	1 (1.9%)	0.197
Nonsmoker	89 (84.8%)	43 (84.3%)	46 (85.2%)	0.649
Current smoker	8 (7.6%)	3 (5.9%)	5 (9.3%)	
Ex-smoker	8 (7.6%)	5 (9.8%)	3 (5.6%)	
Alcohol (Current)	22 (22.9%)	14 (27.5%)	10 (18.5%)	0.276
Type 2 DM	2 (1.9%)	1 (1.9%)	1 (1.9%)	1.000
Hypertension	6 (5.7%)	4 (7.8%)	2 (3.7%)	0.428
Hyperlipidemia	9 (8.6%)	8 (15.9%)	1 (1.9%)	0.014 *
Disease duration (months)	77.5 ± 68.3	79.4 ± 72.6	75.6 ± 64.6	0.778
Medication				
5-ASA	76 (72.4%)	45 (88.2%)	31 (57.4%)	<0.001 *
Steroid	15 (14.3%)	11 (21.6%)	4 (7.4%)	0.051
Immunomodulators	43 (40.7%)	10 (19.6%)	33 (61.1%)	<0.001 *
Biologics	35 (33.7%)	12 (23.5%)	23 (43.4%)	0.032 *
Disease activity				
Remission	68 (64.8%)	31 (60.8%)	37 (68.5%)	0.782
Mild	18 (17.1%)	10 (19.6%)	8 (14.8%)	
Moderate	16 (15.2%)	8 (15.7%)	8 (14.8%)	
Severe	3 (2.9%)	2 (3.9%)	1 (1.8%)	

* *p* < 0.05, IBD: inflammatory bowel disease, 5-ASA: 5-aminosalicylic acid.

**Table 2 nutrients-16-03763-t002:** Nutrition laboratory findings of IBD patients.

Variables	IBD (n = 105)	Ulcerative Colitis (n = 51)	Crohn’s Disease (n = 54)	*p*-Value
Hemoglobin (Female: 12.1–15.1, Male: 13.8–17.2 g/dL)	13.1 ± 1.8	13.2 ± 1.8	13.0 ± 1.9	0.574
Total protein (6.7–8.3 g/dL)	7.1 ± 0.5	7.1 ± 0.4	7.2 ± 0.6	0.436
Albumin (3.8–5.3 mg/dL)	4.1 ± 0.4	4.1 ± 0.3	4.1 ± 0.4	0.888
Total blood calcium (8.4~10.2 mg/dL)	8.9 ± 0.4	8.8 ± 0.4	8.9 ± 0.4	0.981
Iron (43~172 µg/dL)	72.7 ± 44.1	82.8 ± 44.0	63.3 ± 42.5	0.024 *
Ferritin (Female: 24~307 ng/mL, Male: 24~336 ng/mL)	77.7 ± 106.4	63.4 ± 59.2	90.6 ± 135.1	0.197
Phosphate (2.5~4.5 mg/dL)	3.60 ± 0.64	3.53 ± 0.64	3.66 ± 0.65	0.279
Folic acid (3.1~20.5 ng/mL)	8.97 ± 4.50	9.60 ± 3.80	8.38 ± 5.03	0.166
Magnesium (Female 19–2.5, Male 1.8–2.6 mg/dL)	2.11 ± 0.17	2.15 ± 0.19	2.08 ± 0.15	0.036 *
25-OH vitamin D (<20 ng/mL, deficiency)	19.5 ± 10.2	20.1 ± 10.6	19.0 ± 9.9	0.567
Vitamin B 12 (187~883 pg/mL)	661.4 ± 319.6	722.6 ± 327.5	607.1 ± 305.1	0.068
Zinc (66~ 110 µg/dL)	76.4 ± 16.3	77.2 ± 11.5	75.7 ± 20.0	0.631

* *p* < 0.05, IBD: inflammatory bowel disease.

**Table 3 nutrients-16-03763-t003:** Comparison of muscular status of IBD patients.

Variables	IBD (n = 105)	Ulcerative Colitis (n = 51)	Crohn’s Disease (n = 54)	*p*-Value
Skeletal muscle mass (kg)	20.9 ± 5.9	22.7 ± 6.2	19.2 ± 5.5	0.005 *
Muscle mass (kg)	43.3 ± 9.1	44.0 ± 9.5	42.6 ± 8.7	0.444
SMI (%): (SMM/weight) × 100	34.3 ± 5.1	35.8 ± 5.5	32.8 ± 4.7	0.004 *
SMM/Height^2^ (kg/m^2^)	7.6 ± 1.7	8.2 ± 1.9	7.0 ± 1.5	<0.001 *

* *p* < 0.05, UC: ulcerative colitis; CD: Crohn’s disease; IBD: inflammatory bowel disease; SMI: skeletal muscle index; SMM: skeletal muscle mass.

## Data Availability

The datasets generated or analyzed during this study are available from the corresponding author on reasonable request.

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
