# Peer review of "Micronutrient Deficiency and Muscular Status in Inflammatory Bowel Disease"

_nutrients, 2024, doi:10.3390/nu16213763_

Round 1
Reviewer 1 Report
Comments and Suggestions for Authors
The authors have analyzed the concentration of micronutrients and bioelectrical impedance analyses in a cross-sectional study of patients with IBD. The patient cohort consisted of 105 patients that were prospectively enrolled on Jeju Island in South Korea over a two-year period.
I have the following major concerns:
1) The study is limited by the relatively small sample size and it could also be limited by patient selection and may lack novelty. It is well known that CD patients have more nutritional problems than UC patients. The study also lacks a healthy non-IBD control group from the same population for comparison of body composition and micronutrient concentrations. The body composition data may be relatively novel compared to the rest of the results, but comparison to other studies that have measured the same variables should be done.
2) In the methods section there should be some information about patient selection in order to assess the influence of patient selection on the results. How many patients were included relative to the total number of patients with IBD in the study period? The major difference in age between CD and UC patients suggest that patient selection may explain differences in many variables.
3) There is no information about how all analytes were measured, please add some details. If all analyses were done as a part out routine at a hospital, some more information should still be presented.
4) In the methods section, please define hyperlipidemia. Similarly “Alcohol status” presented in table 1 and in the manuscript body should not be a dichotomous variable without definition. Is medication current or ever used?
Minor comments
Age and BMI were analyzed as if they had normal distribution. Please test and confirm that this was done in the manuscript or change to a non-parametric test to compare between groups.
In table 1, presentation of both male and female gender (n%) is redundant when used as a dichotomous variable.
In table 2, the normal values seem to lack information. Hemoglobin values usually have decimals. Is calcium ionized or free? Ferritin normal range differs between men and women, please adjust
Line 70. A parenthesis after ASA is missing?
Line 92. The “hight.16” is perhaps a reference? Please check and correct.
Line 109: please rephrase the statement about alcohol history.
Table 1: When comparing categorical data with less than 5 observations in one or more squares and exact test (e.g. Fischers) is recommended, not chi square. E.g. family history, DM2, hypertension, hyperlipidemia.
Line 157: It is stated that vitamin D deficiency predisposes to sarcopenia. Please do not suggest that this is a causal relationship and change the “associated with” or similar.
Line 160-62: Are there studies that have found that vitamin D supplementation affects disease activity in patients with CD? Are the studies not only observational? Please clarify in the manuscript and adjust the statements. Vitamin D supplementation should of course be used for several other reasons including bone health.
Line 165: Reference 21 does not examine whether sarcopenia is an independent risk factor for surgery. Please change the sentence or add another reference.
Line 173: The sentence “ In patients with CD, there was one severe case (1.8%) due to the use of immunomodulators and biological agents” is difficult to understand. What is an severe case and it can not be stated that the disease was severe due to the use of medication. Please adjust.
Comments on the Quality of English LanguageMinor issues, please proof-read and see comments.
Author Response
Thank you for providing such valuable feedback despite your busy schedule.
1) The study is limited by the relatively small sample size and it could also be limited by patient selection and may lack novelty. It is well known that CD patients have more nutritional problems than UC patients. The study also lacks a healthy non-IBD control group from the same population for comparison of body composition and micronutrient concentrations. The body composition data may be relatively novel compared to the rest of the results, but comparison to other studies that have measured the same variables should be done.
Thank you for the valuable review. We are planning a multi-center study with other hospitals to obtain a larger sample size and will begin enrolling patients early next year. The next study will also include a healthy control group.
2) In the methods section there should be some information about patient selection in order to assess the influence of patient selection on the results. How many patients were included relative to the total number of patients with IBD in the study period? The major difference in age between CD and UC patients suggest that patient selection may explain differences in many variables.
Methods: We appreciate the reviewer’s comments. We have added information on the total number of IBD patients during the study period and provided details on the exclusion criteria for the excluded patients.
3) There is no information about how all analytes were measured, please add some details. If all analyses were done as a part out routine at a hospital, some more information should still be presented.
Thank you for the suggestion. We have revised the methods section to provide a more detailed explanation of how the substances in the blood were analyzed.
4) In the methods section, please define hyperlipidemia. Similarly “Alcohol status” presented in table 1 and in the manuscript body should not be a dichotomous variable without definition. Is medication current or ever used?
Variable Definitions: Thank you for pointing this out. We have added clearer definitions for the terms used in the paper.
Minor comments
[1] Age and BMI were analyzed as if they had normal distribution. Please test and confirm that this was done in the manuscript or change to a non-parametric test to compare between groups.
Thank you for the feedback. We have re-evaluated this aspect as per the reviewer's suggestion and added comments regarding this in the manuscript.
[2] In table 1, presentation of both male and female gender (n%) is redundant when used as a dichotomous variable.
Thank you for the suggestion. We have revised the presentation of gender to be simpler and more intuitive.
[3] In table 2, the normal values seem to lack information. Hemoglobin values usually have decimals. Is calcium ionized or free? Ferritin normal range differs between men and women, please adjust
Normal Values: Thank you for the feedback. We have adjusted the values in Table 2 according to the reviewer’s recommendations.
[4] Line 70. A parenthesis after ASA is missing?
Thank you for pointing this out. We have added the missing parenthesis after "ASA."
[5] Line 92. The “hight.16” is perhaps a reference? Please check and correct.
Thank you for the suggestion. We have corrected the reference accordingly.
[6] Line 109: please rephrase the statement about alcohol history.
Thank you for the feedback. We have clarified the definition of alcohol consumption and reflected this in the manuscript.
[7] Table 1: When comparing categorical data with less than 5 observations in one or more squares and exact test (e.g. Fischers) is recommended, not chi square. E.g. family history, DM2, hypertension, hyperlipidemia.
Categorical Data Analysis (Table 1): Thank you for the comment. We have addressed the issue as per the reviewer’s suggestion.
[8] Line 157: It is stated that vitamin D deficiency predisposes to sarcopenia. Please do not suggest that this is a causal relationship and change the “associated with” or similar.
Vitamin D and Sarcopenia (Line 157): Thank you for the feedback. We have modified the expression in line with the reviewer’s recommendation.
[9] Line 160-62: Are there studies that have found that vitamin D supplementation affects disease activity in patients with CD? Are the studies not only observational? Please clarify in the manuscript and adjust the statements. Vitamin D supplementation should of course be used for several other reasons including bone health.
Vitamin D Supplementation (Lines 160-162): Thank you for the suggestion. We have revised the wording as per the reviewer’s comment.
[10] Line 165: Reference 21 does not examine whether sarcopenia is an independent risk factor for surgery. Please change the sentence or add another reference.
Reference 21 (Line 165): Thank you for the feedback. We have changed the reference according to the reviewer’s suggestion.
[11] Line 173: The sentence “ In patients with CD, there was one severe case (1.8%) due to the use of immunomodulators and biological agents” is difficult to understand. What is an severe case and it can not be stated that the disease was severe due to the use of medication. Please adjust.
Severe Case Description (Line 173): Thank you for pointing this out. We have modified the content in the manuscript as per the reviewer’s recommendation.

Reviewer 2 Report
Comments and Suggestions for Authors
-- Use oxford comma in the whole text
-- “5-aminosalicylic acid (ASA”
Close the parenthesis
- - “Continuous variables were described as mean ± standard deviation”
Did you test for normal distribution?
- - The percentages of drug use were currently or in the patients’ history?
- - “Bioelectrical impedance analysis indicated that skeletal muscle was lower in patients 133 with CD compared to UC”
Was there correlation with disease activity, current drugs, BMI, etc.?
Comments on the Quality of English Languagegood
Author Response
Thank you for providing such valuable feedback despite your busy schedule.
[1] Use oxford comma in the whole text
We appreciate your valuable comments. We changed in manuscript.
[2] “5-aminosalicylic acid (ASA”
Close the parenthesis
Thank you for pointing this out. We have added the missing parenthesis after "ASA."
[3] “Continuous variables were described as mean ± standard deviation”
Did you test for normal distribution?
Thank you for the feedback. We have adjusted the values in Table 2 according to the reviewer’s recommendations.
[4] The percentages of drug use were currently or in the patients’ history?
Thank you for the suggestion. We only enroll current patient who take medication in study period.
[5] “Bioelectrical impedance analysis indicated that skeletal muscle was lower in patients 133 with CD compared to UC”
Thank you for the feedback. We have re-evaluated this aspect as per the reviewer's suggestion and added comments regarding this in the manuscript. This may will be reflect our next study.
[6] Was there correlation with disease activity, current drugs, BMI, etc.?
Thank you for the feedback. Actually we didn’t postulate any meaningful result in those variables. We will conduct next other multicenter trial in this topics. This may will be reflect our next study.

Round 2
Reviewer 1 Report
Comments and Suggestions for Authors
The manuscript has been through major revision. Perhaps the most important changes have bee in the Methods section and section 2.2. in particular. The limitations have been mentioned in the discussion and there is a need for larger studies with a prospective design.
Comments on the Quality of English LanguageThe very last sentence "managing malnutrition in IBD in prospective setting" is difficult to understand and may be adjusted during final proof-reading? Missing "a" or other word?
Author Response
We appreciate the reviewer's accurate observations. We have revised the sentence to convey a clearer meaning.
Reviewer 2 Report
Comments and Suggestions for Authors
- "We analyzed patients' age, gender, smoking and alcohol 114 status, family history, past medical history, body mass index (BMI), date of IBD diagnosis, 115 disease activity, and medical treatment in terms of averages and proportions. Continuous 116 variables were described as mean ± standard deviation, and comparisons between two 117 groups were made using Student's t-test. Non-continuous variables were analyzed using 118 the chi-square test. All variables, except family history, diabetes, hypertension, and hy-119 perlipidemia, followed a normal distribution."
Categorical variables cannot have normal or not normal distribution. Only continuos varbiales are normally or not normally distributed
- "The percentages of drug use were currently or in the patients’ history?
Thank you for the suggestion. We only enroll current patient who take medication in study period."
I asked another thing. Regarding the percentages of drug use, are you reporting a drug only if it is currently used or even if the drugs has been used in the past from the patient?
- "Bioelectrical impedance analysis indicated that skeletal muscle was lower in patients 133 with CD compared to UC
Was there correlation with disease activity, current drugs, BMI, etc.?"
You did not asnwer to my question
Comments on the Quality of English Languageok
Author Response
Q 1. "We analyzed patients' age, gender, smoking and alcohol status, family history, past medical history, body mass index (BMI), date of IBD diagnosis, disease activity, and medical treatment in terms of averages and proportions. Continuous variables were described as mean ± standard deviation, and comparisons between two groups were made using Student's t-test. Non-continuous variables were analyzed using the chi-square test. Categorical variables cannot have normal or not normal distribution. Only continuos varbiales are normally or not normally distributed
A 1. We appreciate the reviewer's precise feedback. We have removed the ambiguous expression.
Q2. "The percentages of drug use were currently or in the patients’ history?
Thank you for the suggestion. We only enroll current patient who take medication in study period." I asked another thing. Regarding the percentages of drug use, are you reporting a drug only if it is currently used or even if the drugs has been used in the past from the patient?
A2.
Thank you for pointing out the shortcomings in our paper. First, the definition of "drug" we intended in our study refers to IBD (Inflammatory Bowel Disease) treatments. We excluded all patients who had received IBD treatments prior to the study period. To convey our intention more accurately, we have modified the exclusion criteria.
Q3. - "Bioelectrical impedance analysis indicated that skeletal muscle was lower in patients with CD compared to UC
Was there correlation with disease activity, current drugs, BMI, etc.?"
You did not asnwer to my question
A3. Thank you for pointing out the aspects that we analyzed but failed to mention in the paper. In this study, no correlation was found. However, we believe that significant results could be obtained in a study with a larger sample size by controlling for disease activity, current drugs, and BMI separately, and attempting univariate and multivariate regression analyses.
